# Earlier Detection of Alzheimer’s Disease Based on a Novel Biomarker *cis* P-tau by a Label-Free Electrochemical Immunosensor

**DOI:** 10.3390/bios12100879

**Published:** 2022-10-17

**Authors:** Ayoub Shiravandi, Farzaneh Yari, Nahid Tofigh, Mohammad Kazemi Ashtiani, Koorosh Shahpasand, Mohammad-Hossein Ghanian, Faezeh Shekari, Farnoush Faridbod

**Affiliations:** 1Department of Cell Engineering, Cell Science Research Center, Royan Institute for Stem Cell Biology and Technology, Academic Center for Education, Culture and Research (ACECR), Tehran 1665659911, Iran; 2Center of Excellence in Electrochemistry, School of Chemistry, College of Science, University of Tehran, Tehran P.O. Box 14155-6455, Iran; 3Laboratory of Neuro-Organic Chemistry, Institute of Biochemistry and Biophysics (IBB), University of Tehran, Tehran 1417935840, Iran; 4Department of Stem Cells and Developmental Biology, Cell Science Research Center, Royan Institute for Stem Cell Biology and Technology, Academic Center for Education, Culture and Research (ACECR), Tehran 1665659911, Iran; 5Advanced Therapy Medicinal Product Technology Development Center, Royan Institute for Stem Cell Biology and Technology, Academic Center for Education, Culture and Research (ACECR), Tehran 1665659911, Iran

**Keywords:** Alzheimer’s disease, early detection, antibody, *cis* P-tau, human serum, electrochemical biosensor

## Abstract

Early detection of *cis* phosphorylated tau (*cis* P-tau) may help as an effective treatment to control the progression of Alzheimer’s disease (AD). Recently, we introduced for the first time a monoclonal antibody (mAb) with high affinity against *cis* P-tau. In this study, the *cis* P-tau mAb was utilized to develop a label-free immunosensor. The antibody was immobilized onto a gold electrode and the electrochemical responses to the analyte were acquired by electrochemical impedance spectroscopy (EIS), cyclic voltammetry (CV), and differential pulse voltammetry (DPV). The immunosensor was capable of selective detection of *cis* P-tau among non-specific targets like *trans* P-tau and major plasma proteins. A wide concentration range (10 × 10^−14^ M–3.0 × 10^−9^ M) of *cis* P-tau was measured in PBS and human serum matrices with a limit of detection of 0.02 and 0.05 pM, respectively. Clinical applicability of the immunosensor was suggested by its long-term storage stability and successful detection of *cis* P-tau in real samples of cerebrospinal fluid (CSF) and blood serum collected from human patients at different stages of AD. These results suggest that this simple immunosensor may find great application in clinical settings for early detection of AD which is an unmet urgent need in today’s healthcare services.

## 1. Introduction

Alzheimer’s disease (AD) is a chronic, devastating dysfunction of neurons in the brain characterized by progressive memory impairment, noticeable personality changes, cognitive impairment, dementia, and eventual death [1]. AD represents one of the biggest medical challenges over the past five decades [1,2,3,4]. According to the World Health Organization (WHO), more than 50 million people suffer from dementia and this number is expected to exceed 152 million by the year 2050 [5,6]. One of the biggest challenges in control of AD is the early diagnosis of disease which is crucial for effectiveness of the drugs and subsequently less nerve damage [4,7,8,9,10]. By early detection of AD in the first stages of the disease, the routine treatments can be more effective to prevent disease progression and reduce the mortality and healthcare costs [9,10].

AD is considered a continuum of gradual brain changes progressing over about 25 years before the first symptoms appear [11]. In the early stages of AD, with the low concentration of biomarkers, the AD is a mild cognitive impairment (MCI), but over time and as the amount of disease-causing agents increases, AD appears with all its associated symptoms [12]. Therefore, AD has the potential to be diagnosed before the dementia stage. The advanced tools of AD diagnosis are still incapable of early detection, mainly due to the lack of a biorecognition element for the earlier biomarkers of AD. Currently, the most well-known biomarkers that are agreed upon are the tau proteins and amyloid-β peptides. These biomarkers are identified in cerebrospinal fluid (CSF) samples, which are commonly used for AD diagnosis in clinical practice [10,13]. In this regard, a variety of biosensors have been developed in the literature for more accurate and rapid diagnosis of AD based on these biomarkers [14,15,16,17,18,19,20,21,22,23,24,25,26,27,28,29,30]. However, none of these biomarkers is the exclusive early driver of the disease which can be meaningfully detected at early stages of the disease. Hence, there is an unmet need for a biosensor based on an early stage biomarker of AD.

Tau is a microtubule-related protein that contributes to stabilizing microtubules in neural cells [22]. The abnormal phosphorylation process of tau proteins is an essential characteristic of tau pathology in AD [31] because the hyperphosphorylated tau (P-tau) proteins aggregate in the form of paired helical filaments which form intracellular neurofibrillary tangles (NFTs) in the dendritic spines of neurons [32]. By increasing and deposition of NFTs, the normal function of nerve cells is disrupted which eventually leads to cell death [33,34]. The P-tau protein exists in two conformations, *trans* and *cis*, of which the latter has been shown to be extremely neurotoxic and more prone to accumulation [35,36,37,38]. Clinical data shows that in the early stages of AD, an increase in the concentration of the *cis*, but not *trans*, isoform of P-tau appears in the brains of humans with MCI [39,40,41,42]. In addition, the concentration of *cis* P-tau increases more severely than other pathogenic species of tau tangles in AD [38,43,44,45]. This robust *cis* P-tau pathology (cistauosis) has been also seen after traumatic brain injury in humans and mice which is the best-known environmental risk factor for AD. These data show that the cistauosis appears long before other tauopathy factors and can be considered as a precursor of tauopathy and a specific signal to predict AD at the earlier stages [36,39,41,46,47,48,49]. However, no biosensor has been developed for AD detection based on cistauosis. Particularly, cis pT231-tau is a central mediator and an early driver of neurodegeneration upon tauopathy process. Moreover, it has a prion nature and spreads in the CSF as well as blood. Thus, we herein employed a monoclonal antibody against the pathogenic P-tau species to detect and even predict neurodegeneration through a non-invasive procedure using blood samples.

The cut-off value of P-tau in CSF is considered in the range of 10 to 20 pM [50]; however, the serum level of P-tau is expected much lower (~20 times) because P-tau levels in brain is amplified by spilling of intracellular tau from dying neurons in the central nervous system which are in direct contact with CSF [51]. As a result, the concentration of P-tau in CSF is proportional to the amount of released protein from the brain neurons, while the P-tau content in serum is determined by the amount of protein released from CSF into serum, which is limited by the relatively large size of P-tau that limits its permeability through the blood–brain barrier [52]. Due to the extremely low levels of P-tau in serum, current assay methods cannot be used to detect P-tau in blood samples [53]. In addition, the serum concentration of *cis* P-tau is further lower than that of the total P-tau due to co-existence of the two conformations (*cis* and *trans*) of P-tau in human serum [35,36], and hence a highly sensitive measurement is required to detect *cis* P-tau in serum [54,55,56].

Recently, we developed a monoclonal antibody (mAb) against the *cis* isoform of P-tau protein, which showed selectivity to block this driver of AD and delayed the progression of AD in mice [39]. Thereafter, we established a *cis* P-tau mAb with high affinity against the human samples [53]. This proprietary human *cis* P-tau mAb has been exploited in this work to develop an immunosensor for earlier detection of AD based on cistausis. To establish an easy-to-use biosensor platform, the antibody was immobilized on the surface gold electrodes, and the signal changes upon analyte addition were acquired by conventional electrochemical analyses. This label-free detection mechanism allows for easy, fast, and cost-effective analysis without the need for labeling the analyte with the signal tags (e.g., enzymes and metal nanoparticles).

Our result shows applicability of this simple biosensor in clinical settings for early detection of AD.

## 2. Materials and Methods

### 2.1. Materials

Briefly, 3-Mercaptopropionic acid (MPA, 99%), 11-Mercaptoundecanoic acid (MUA, 98%), N-(3-dimethyl amino propyl)-N′ ethyl carbodiimide hydrochloride (EDC), N-hydroxysuccinimide (NHS), bovine serum albumin (BSA), potassium ferricyanide (K_3_Fe(CN)_6_), potassium ferro cyanide (K_4_Fe(CN)_6_), sodium hydroxide (NaOH), phosphate buffered saline (PBS: 10 mM Na_2_HPO_4_, 10 mM KH_2_PO_4_, 150 mM NaCl, pH 7.5), potassium chloride (KCl), ethanol 99%, HRP-conjugated goat anti-mouse IgG antibody, beta-2-microglobolin, immunoglobulin g (≥95%), hemoglobin, 3,3′,5,5′-tetramethylbenzidine (TMB), and sulfuric acid (H_2_SO_4_, 99%) were purchased from Sigma-Aldrich. Human albumin (20%) was purchased from Biotest Pharma GmbH (Germany). Alumina powder (5.0, 0.30, and 0.1 μm) was purchased from Metsuco (Houston, USA) and Sigma-Aldrich (Bengaluru, Karnataka, India). Yellow lumbar puncture needle (20-gauge) was purchased from Millipore (Frankfurter, Germany). Screen-printed gold electrodes (SPGE) (Aux.: Pt; Ref.: Ag)/Ink AT and gold disk electrodes (GDE) were purchased from Metrohm (Herisau, Switzerland) and Azar Electrode (Urmia, Iran), respectively. Ultrapure deionized water (Milli-Q, 18.2 MΩ cm) was obtained from the Milli-Q system and used in the experiments and washing.

### 2.2. Electrode Preparation

For the purpose of removing and cleaning any impure materials from the GDE surface, first, the bare electrodes were polished with alumina powders (5.0, 0.30, and 0.1 μm) for 5 min with each of the sizes, subsequently immersed in piranha solution (1:3, 30% H_2_O_2_ and concentrated 98% H_2_SO_4_) for 5 min, and then rinsed with deionized water. To achieve a maximum cleaning, the GDE was cleaned by electrochemical method in 0.5 M NaOH solution via cyclic voltammetry (CV) at 0.1 V/s scan rate and in the −0.3 to 0.8 V potential range. Then electrodes were immersed into 0.5 M H_2_SO_4_ and CV conducted at 0.1 V/s scan rate and in the −0.3 to 0.8 V potential range. After the processes, the GDE was washed with deionized water and dried.

### 2.3. Fabrication of Immunosensor

*Cis* P-tau mAb, the same clone as #113 from Creativebiolabs, was produced with ~99% purity in the lab according to the procedure of our previous publications [57]. The immunosensor was fabricated by covalent immobilization of the antibody on the surface of the gold electrode (Figure 1). Two thioacid cross-linkers (MPA and MUA) were used to establish attachment sites on the gold for antibody immobilization. A mixture of MUA/MPA (70/30) in ethanol (10 mM) was dropped on the gold surface and dried for 8 h at room temperature (RT). The linker-modified electrode was washed by ethanol to remove all non-bonded molecules. Then, a mixture of EDC (20 mM)/NHS (5 mM) in 0.1 M PBS (pH = 5.5–6) was dropped onto the surface at RT and left to dry. Next, a PBS solution of *cis* P-tau mAb (10 μg/mL, pH 7.5) was dropped onto surface and kept overnight at 4 °C to dry and then rinsed with PBS solution. In the last stage, to block non-specific adsorption, a PBS solution of BSA (1% *w*/*v*) was dropped onto the electrode surface, kept for1 h, and rinsed with PBS. The immunosensors based on SPGE were fabricated via a similar protocol, without need for the initial electrode cleaning. The fabricated immunosensors were stored at 4 °C until use.

### 2.4. Morphological Characterization of Immunosensor Surface

In each step of the immunosensor assembly, the surface morphologies of the electrodes were analyzed over a 10 × 10 μm area by atomic force microscopy (AFM; VEECO CP II, Veeco Instruments Inc., New York, NY, USA). The imaging process was operated in contact mode, and scanned at a rate of 1.0 Hz. To show the capability for *cis* P-tau interaction with the immobilized antibody, the modified electrode was subjected to incubation with the *cis* P-tau solution in PBS (1.0 × 10^−12^ M) for 30 min at RT, followed by washing with 5 mL of PBS before imaging.

### 2.5. Electrochemical Characterization of Immunosensor

Electrochemical impedance spectroscopy (EIS) and CV were used to monitor the electrochemical performances of immunosensor at different steps of the layer-by-layer procedure of immunosensor fabrication. The electrochemical measurements were conducted in 10 mM K_4_Fe(CN)_6_/K_3_Fe(CN)_6_ (1:1 ratio) by using a three-electrode set up including the modified Au electrode (1.5 mm^2^ surface area) as working electrode (WE), a Pt wire as an auxiliary electrode (AE), and Ag/AgCl as the reference electrode (RE). The EIS and CV were carried out using an EC-Lab (Bio-Logic, sp-200, Seyssinet-Pariset, France) and a CHI 660C potentiostat (CH Instruments Inc., Austin, TX, USA), respectively. For EIS analysis, a bias voltage of 10 mV was applied between WE and CE over a frequency range of 0.1–1000 Hz. The generated EIS data were fitted to the Randles′ equivalent circuit model using ZView software (Solartron Analytical, Farnborough, UK).

### 2.6. Measurement of cis P-tau in PBS and Human Serum

To investigate the electrochemical response of fabricated immunosensor to the different concentrations of *cis* P-tau, the differential pulse voltammetry (DPV) was conducted in 10 mM K_4_Fe(CN)_6_/K_3_Fe(CN)_6_ (1:1 ratio) in PBS at pH 7.5. Different solutions of *cis* P-tau in PBS or undiluted human serum with successive concentrations (0 × 10^−14^ M to 3.0 × 10^−9^ M) were prepared. The human serum was collected from clotted normal blood and was used fresh. Prior to the analysis, each *cis* P-tau solution was dropped onto the WE surface, incubated at RT for 30 min, and subsequently washed with PBS.

### 2.7. Preparation of Real Human Samples

The written informed consent was obtained from all patients and the sample collection protocol was approved by the Ethics Committee of Royan Institute. After the initial survey including mini-mental state examination (MMSE) evaluation, patients underwent CSF sampling. The CSF samples were drained using 20-gauge yellow lumbar puncture needle inserted into the subarachnoid space at the L3–4 or L4–5 interspace after sterile preparation. The 2 to 5 mL of CSF was then collected in sterile plastic tubes for evaluation of biomarker (CSF *cis* P-tau) using an in-house ELISA and the immunosensor. Similarly, for the control group, the CSF sample was drained using the same method at the time of spinal anesthesia and subjected for evaluation of *cis* P-tau. Blood samples were collected from the AD patients at different stages and healthy controls in collaboration with Sasan hospital of Tehran, Iran. Serum samples were derived from the clotted blood and were analyzed fresh. The analysis was carried out blindly according to the clinical status.

### 2.8. Measurement of cis P-tau in Real Samples

The CSF samples were diluted 1/100 in 0.1 M PBS prior to analysis and the serum samples were analyzed without any pretreatment. For immunosensor measurement, each sample was dropped onto the WE surface, incubated at RT for 30 min, washed with PBS, and then, DPV was conducted in 10 mM K_4_Fe(CN)_6_/K_3_Fe(CN)_6_ (1:1 ratio) in PBS at pH 7.5. To compare with the immunosensor, an in-house ELISA was produced for the detection of *cis* P-tau in real human CSF and serum samples. In brief, 2 μg of anti cis-phosphorylated Thr231-Tau diluted in 100 μL of PBS (0.01 M, pH 7.2) and was immobilized in 96-well ELISA plates and stored at 4 °C overnight. To reduce nonspecific binding, wells were rinsed with 5 × 300 μL Tris-buffered saline/Triton X-100 (TBST) using an automatic plate washer and blocked for 45 min at 37 °C with a blocking solution (TPBS + 0.5 mM BSA). Following additional automated washing, 100 μL/well of serial standards or human serum were added and incubated at 37 °C for 1 h. Following an additional wash step, 100 μL of HRP-conjugated goat anti-mouse IgG diluted 1:500 in BSA was added to each well, incubated for 1 h at 37 °C and washed. The colorimetric reaction was initiated upon the addition of 200 μL of ready-to-use TMB substrate (Seramun Diagnostica GmbH, Germany), and the plates were allowed to rest for 20 min at 37 °C in the dark. Finally, the reaction was stopped with 100 μL/well of 2 N sulfuric acid, and the absorbance of the samples was determined on a microplate reader at 450 nm (Thermo Fisher Scientific, Waltham, MA, USA). The standard curve was prepared by serial dilution of cis P-tau peptide (cis-phosphorylated Thr231- Dmp-Tau (KVAVVRpT (5,5-dimethyl-L-proline) PKSPS) in PBS ranging from 0.0001 μg/mL to 3 μg/mL. The ELISA assay was performed in triplicate experiments and compared with the results obtained using the immunosensor.

### 2.9. Statistical Analysis

All statistical analyses were done using Student′s *t*-test and *p* < 0.05 was considered to be statistically significant.

## 3. Results and Discussion

### 3.1. Fabrication and Characterization of Immunosensor

The topographical changes of the electrode surface during the multi-step procedure of the immunosensor fabrication were monitored using AFM. Figure 2 shows the AFM 3D images at the consequent stages of the sensor fabrication. After deposition of the NHS-activated thiol linkers on the GE, the roughness of the electrode surfaces decreased slightly from 142 nm to 135 nm (Figure 2A,B). Upon addition of the macromolecular agent of the *cis* P-tau antibody, the surface roughness increased dramatically to 144 nm (Figure 2C), which is similar to that observed in other studies [58,59]. Moreover, the addition of analyte, *cis* P-tau produced a small change in the surface roughness from 144 nm to 148 nm (Figure 2D). The AFM data could confirm the successful deposition of the agents at each stage.

EIS and CV were used to confirm the layer-by-layer deposition of substances on the gold electrode surface during the multi-step fabrication process (Figure 3). EIS is a powerful instrument for surface characterization and monitoring changes to the interfaces. The EIS data were showed through Nyquist complex-plane diagrams (Figure 3A). In these diagrams, the semicircle diameter of each impedance spectrum represents the interfacial charge-transfer resistance (R_ct_) at each step of the immunosensor construction. At the first step, chemical modification of gold surface with MUA/MPA resulted in a remarkable increase in the R_ct_ (1.53 to 7.08 kΩ), which could be related to the lower surface concentration of the probe due to the existence of long carbon chains in the MUA-MPA layer, which could limit the diffusion of Fe(CN)_6_ ^3−/4−^ redox couple towards the electrode surface. Additionally, the peak shift is an indication of the enhancement of difficulty of electron transfer, which was probably caused by the negative charge of the electrode surface via the formation of R-COO^−^ layer. This behavior was realized by a remarkable decrease of the current intensity in CV diagram of the electrode modified with MUA/MPA compared to the bare electrode (Figure 3B). Subsequently, treatment of the MUA/MPA-modified electrode by EDC/NHS showed a declined R_ct_ (2.16 kΩ) and an increased current intensity in CV (Figure 3B). This could contribute to the elimination of the negative charges through NHS ester formation on the carboxylic groups, which generally reduces the negative charges of the electrode surface and facilities the probe diffusion ability. These results confirmed that the CO_2_^-^ groups of MUA/MPA were activated with NHS coupling. Upon the immobilization of *cis* P-tau mAb on the activated gold electrode, a remarkable increase in the R_ct_ (3.28 kΩ) and decrease in the current intensity (Figure 3B) were observed due to the binding and accumulation of the macromolecular antibody on the electrode surface which may cause more blockage in the diffusion path of the probe ions towards the electrode surface. The results of EIS and CV were in agreement and confirmed the success of the layer-by-layer assembly process during fabrication and performance of the immunosensor. Upon addition of *cis* P-tau solution onto the electrode surface, the peak current was further decreased, suggesting the successful recognition of *cis* P-tau by the antibody and accumulation of the *cis* P-tau macromolecules on the electrode which can hinder electron transfer between the electrode surface and the solution containing the [Fe(CN)_6_]^3−/4−^ redox probe. All the measurements were conducted under pH 7.5, which was determined as the optimal pH of medium for maximal electrochemical signal production in response to *cis* P-tau (Appendix A).

### 3.2. Selectivity of Immunosensor

To investigate the selectivity of the immunosensor to *cis* P-tau, the possibility of an off-target response to *trans* P-tau and major plasma proteins including albumin, beta-2-microglobolin, immunoglobulin G, and hemoglobin was studied by DPV analysis (Figure 4). These proteins were chosen as major off-targets because trans P-tau is the other isomer of P-tau, which is not considered as early driver of AD, albumin and beta-2-microglobolin are abundant in CSF and serum and beta-2-microglobolin is routinely tested as a representative of the heavy inflammatory marker proteins in CSF, IgG is an immunoprotein that is elevated in CSF in cases of infection, and hemoglobin is routinely tested as a blood protein [60,61,62,63,64]. The presence of these components in biological fluids may interfere with detection of low abundant biomarkers in clinical samples [65]. The immunosensor response to these proteins was analyzed by DPV method (Figure 4A), and the current intensity change upon incubation with each protein was measured from five independent experiments (Figure 4B). The data showed that upon incubation with the off-target proteins, the recorded signal was similar to the blank PBS, while a sharp signal rise was observed with the target protein *cis* P-tau even at lower concentration. The result affirmed that no considerable non-selective binding occurred between anti-*cis* P-tau antibody and the off-target proteins and the produced signal with incubation of *cis* P-tau was the result of specific antibody–antigen interactions. This specificity against the *cis* isomer of P-tau was further confirmed by analysis of signal production in response to higher concentrations of *trans* P-tau (100, 1000, and 2000 pM), which showed no significant signal change at all the levels of this off-target isomer (Appendix A).

### 3.3. Characterization of the Immunosensor Response to cis P-tau in PBS and Human Serum

After exploring and determining various optimal experimental conditions, the analytical performance of this immunosensor was evaluated by DPV measurements. Firstly, the matrix effect on the current response of immunosensor was studied by analyzing different blank matrices, PBS, serum, and CSF. The current responses were acquired upon the addition of *cis* P-tau with successive concentration (5.0 × 10^−14^ M to 3.0 × 10^−9^ M) to PBS or human serum (Figure 5). Upon addition of *cis* P-tau, the peak current decreased compared with the blank in a concentration-dependent manner (Figure 5A,B). The current response suppression at different concentrations of *cis* P-tau was calculated from the DPV analysis and plotted in Figure 5C,D. A linear relationship was found between the current change and the *cis* P-tau concentration at the range of 5.0 × 10^−14^ M to 5.0 × 10^−10^ M (insets in Figure 5C,D), with the correlation coefficients more than 0.99 for both PBS and serum samples. At a SNR ≥ 3, the limits of detection (LOD) were calculated to be 2.0 × 10^−14^ M and 5.0 × 10^−14^ M, for PBS and human serum samples, respectively. The obtained LOD was comparable with the electrochemical sensors which have been developed previously for tau protein [17,29,66]. These observations evidenced high sensitivity of the immunosensor for measurement of *cis* P-tau, either in PBS or a complex matrix such as serum.

### 3.4. Clinical Applicability of the Immunosensor for Analysis of Real Samples

To investigate the applicability of our immunosensor in real clinical settings, we used it to detect *cis* P-tau in CSF and serum samples collected from human patients. The *cis* P-tau in real human CSF of healthy and different stages of AD (MCI and dementia) could be detected by the immunosensor in good agreement with the ELISA assay (Table 1). In addition, the standard addition method was used to investigate the sensitivity of the immunosensor to linear changes of the analyte level in the real sample [67,68]. To this end, *cis* P-tau was added to the real CSF samples with subsequent concentrations and the produced DPV signal was acquired (Appendix A and Appendix A). The added standards were recovered in the range from 97.9 to 106.3% with a high repeatability (RSD < 5.5%), showing the acceptable sensitivity of the immunosensor for detection of small changes of *cis* P-tau level in real samples.

In clinical practice, blood samples are more preferable than the CSF samples that are obtained invasively from the patients, especially when a frequent screening program is desired for early detection among high-risk candidates. Although classic approaches such as ELISA can be established to detect *cis* P-tau in CSF (cut-off ~ 10–20 pM) by using our mAb, their sensitivity and selectivity is much lower to determine the extremely lower levels of P-tau in complex samples of blood serum [53]. In our experiences with the in-house ELISA, serum detection was not possible at any stages of AD (data not shown). Moreover, ELISA is a label-based technique with clinical applicability that is challenging due to the costly and time-consuming step of analyte labeling with the signal tags. However, the label-free electrochemical detection mechanism of the immunosensor allows for easy, fast, and cost-effective analysis which potentiate its clinical applicability. The immunosensor measurement of *cis* P-tau in the serum samples of healthy and Alzheimer’s subjects at different stages of AD successfully detected *cis* P-tau at higher levels in AD samples compared with the healthy samples (Table 2). Notably, the *cis* P-tau was detected at levels correlated with the disease stages which suggests the applicability of the immunosensor for serum detection of AD, even at the earlier stages. This label-free detection mechanism allows for easy, fast, and cost-effective analysis without the need for labeling the analyte with the signal tags that are common in classic methods such as ELISA.

In order to evaluate the stability of the immunosensor during long-term storage in clinical practice, the immunosensors were stored at 4 °C and their responses to *cis* P-tau (10 pM) were recorded over a period of 90 days (Appendix A). The results showed a slight attenuation trend during storage, so that the amount of current produced on day 90 decreased by only 3.5% compared to the first day. Therefore, the immunosensor can support an acceptable shelf-life which is crucial for application in clinical services.

## 4. Conclusions

In this study, we utilized a proprietary antibody against *cis* P-tau to develop a novel electrochemical immunosensor for the detection of *cis* P-tau as an early biomarker of AD. The immunosensor was fabricated by covalent immobilization of anti-*cis* P-tau mAb onto a gold electrode surface by using a combination of NHS-activated thiol linkers. By using DPV electrochemical analysis, the *cis* P-tau was detected in PBS and human serum matrices with subsequent concentrations up to 0.02 and 0.05 pM, respectively. The immunosensor performed successfully in the real situation as the *cis* P-tau was measured in human patient CSF samples in good agreement with an in-house ELISA and recovered precisely in manually spiked CSF samples. More importantly, the immunosensor detection was successful in human serum samples collected from AD patients at different disease stages with a low cut-off (0.05 pM), while the detection was not possible with the ELISA. The focus on a specific isomer of P-tau, which is the major early driver of AD, is the main superiority of this work than the previous works on AD detection based on tau measurement (Table 3). As cistausis is a well-known early driver of tauopathy in several neurodegenerative diseases such as AD, this immunosensor may find a unique clinical application in early diagnostic procedures.

## Figures and Tables

**Figure 1 biosensors-12-00879-f001:**
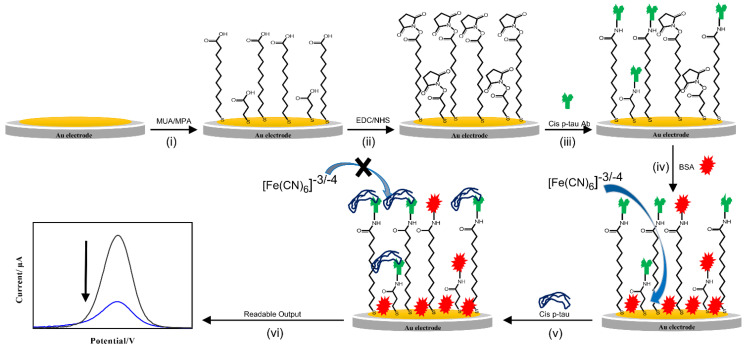
Schematic illustration of immunosensor fabrication and mechanism of performance. (i) Surface modification of gold electrode with the thiol linkers (11-Mercaptoundecanoic acid; MUA and 3-Mercaptopropionic acid; MPA), (ii) activation of the linkers with EDC/NHS, (iii) immobilization of the *cis* P-tau antibody, (iv) blocking of non-specific sites by BSA, (v) addition of the *cis* P-tau and connecting it to the antibody and increasing the resistance of the sensor surface, and (vi) production of a readable signal.

**Figure 2 biosensors-12-00879-f002:**
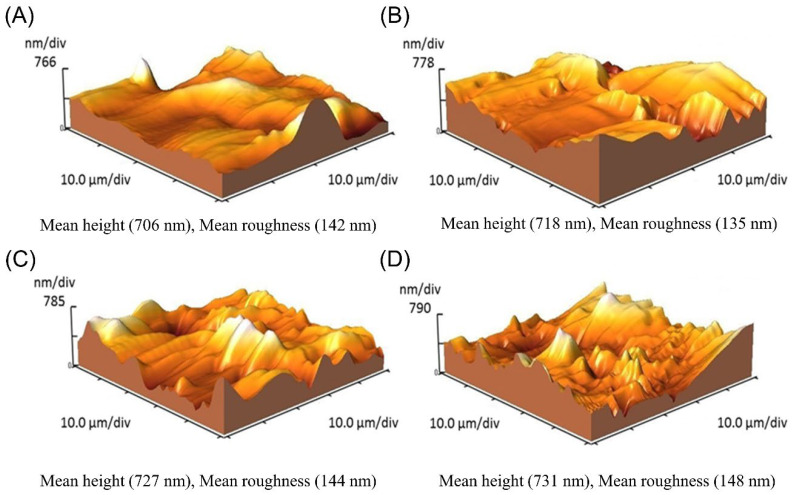
AFM topographic profile of the gold electrode surface in bare state with a roughness of 142 nm (**A**) and functionalized surface by NHS-activated thiol linkers with a roughness of 135 nm (**B**), anti-*cis* P-tau-conjugated surface with a roughness of 144 nm (**C**), and antibody-conjugated surface after the addition of *cis* P-tau with a roughness of 148 nm (**D**).

**Figure 3 biosensors-12-00879-f003:**
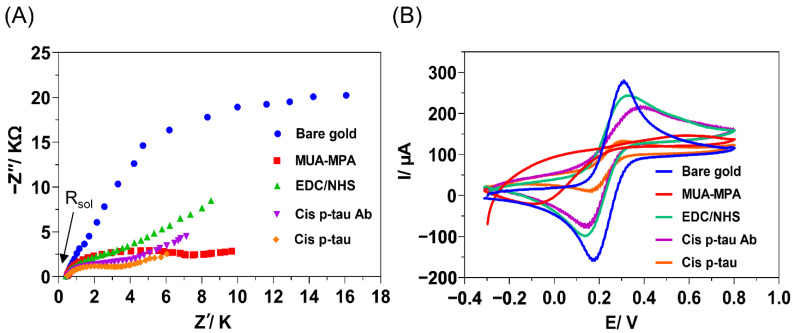
Electrochemical monitoring of the layer-by-layer assembly process carried out for fabrication of immunosensor, including the cleaned bare gold (blue), chemical modification with linkers MUA/MPA (red), activation of linkers with EDC/NHS (green), immobilization of *cis* P-tau mAb (Purple), and deposition of *cis* P-tau (orange). (**A**) Electrochemical impedance spectroscopy (EIS). The imaginary component of EIS (Z″) was plotted against the real component of EIS (Z′) in a Nyquist plot (resistance of solution, (R_sol_ = 33 Ω). (**B**) Cyclic voltammetry (CV) voltammogram. The electrochemical measurements were conducted in PBS (pH 7.5) containing 10 mM K_4_Fe(CN)_6_/K_3_Fe(CN)_6_ (1:1 ratio) by using a three-electrode set up including the modified Au surface as working electrode, a Pt wire as an auxiliary electrode, and Ag/AgCl as the reference electrode.

**Figure 4 biosensors-12-00879-f004:**
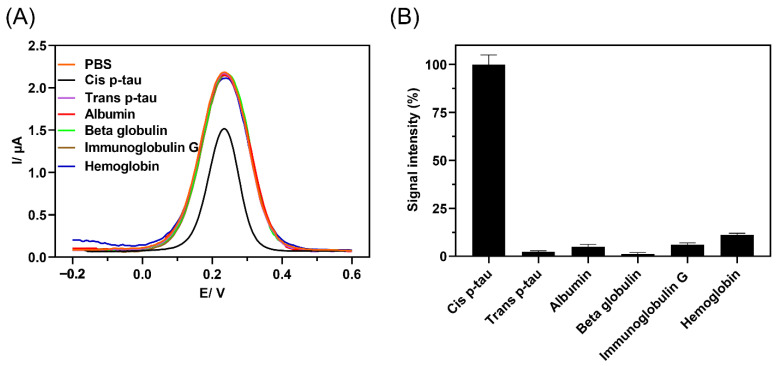
A comparison of DPV response to *cis* P-tau (1.0 × 10^−12^ M) and possible off-targets, including trans P-tau (40 × 10^−12^ M) and major plasma proteins, albumin (1.0 × 10^−9^ M), beta-2-microglobolin (1.0 × 10^−9^ M), immunoglobulin G (IgG) (1.0 × 10^−9^ M), and hemoglobin (1.0 × 10^−9^ M). (**A**) The signal intensity produced by immunosensor in response to protein solutions in PBS containing 10 mM K_4_Fe(CN)_6_/K_3_Fe(CN)_6_ (1:1 ratio) was analyzed by DPV method and (**B**) the immunosensor specificity was measured by relative signal intensity with respect to the baseline (%I_R_), %I_R_ = (I_Bare_ − I_Protein_)/I_Bare_ × 100, where I_Bare_ represents the current intensity of the blank immunosensor and I_Protein_ represents the current intensity of the immunosensor upon incubation with each protein (*n* = 5). The larger the relative current signal difference (%I_R_), the greater the protein recognition.

**Figure 5 biosensors-12-00879-f005:**
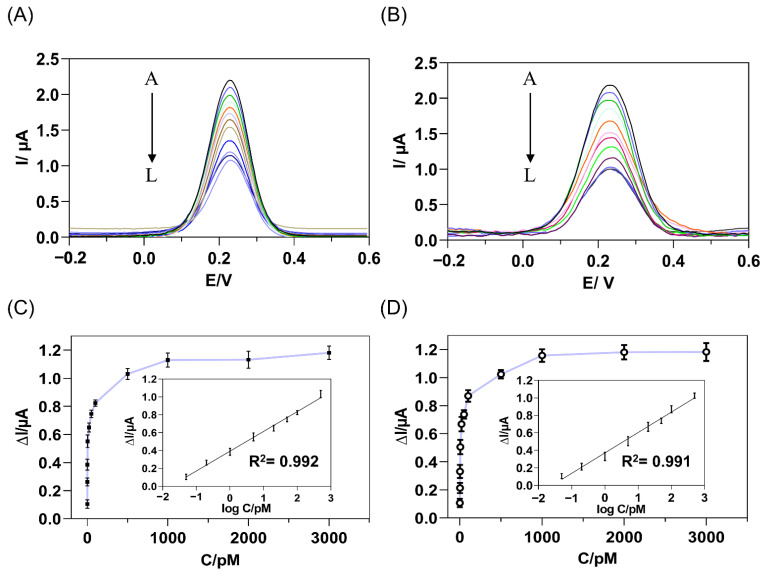
DPV response of the immunosensor to the increasing concentrations of target *cis* P-tau, from A to L: 0, 0.05, 0.2, 1, 5, 20, 50, 100, 500, 1000, 2000, 3000 pM, added to PBS (**A**) or human serum (**B**) containing the [Fe(CN)_6_]^3−/4−^ as a redox probe. The relationship between the current response suppression and *cis* P-tau concentration was calculated from the DPV analysis of PBS (**C**) and human serum (**D**) samples. The insets show the linear calibration curve of the DPV current decrement versus the logarithm of complementary *cis* P-tau concentration (**D**) and the error bars represent standard deviations of the measurements (*n* = 3). The regression equation was obtained as ΔI (μA) = 0.2136 + 0.4042 log C (pM) and ΔI (μA) = 0.23326 + 0.3713 log C (pM) for PBS and human serum samples, respectively, where ΔI is the current decrement and C is the concentration of *cis* P-tau.

**Table 1 biosensors-12-00879-t001:** Determination of *cis* P-tau in real human CSF samples collected from healthy and diseased subjects by using immunosensor and an in-house ELISA.

Sample	*cis* P-tau Concentration (pM)				*p*-Value ^c^
	ELISA ^a^	Immunosensor ^a^	ELISA ^b^	Immunosensor ^b^	
Healthy	15.3 ± 1.2	15.0 ± 0.8	13.9–16.7	14.1–15.9	0.57
MCI	60.3 ± 2.0	59.0 ± 2.2	58.0–62.6	56.5–61.5	0.63
Dementia	80.7 ± 1.7	82.3 ± 1.24	78.7–82.6	80.9–83.7	0.21

^a^ Mean ± standard deviation (*n* = 3); ^b^ confidence interval at 95% level; ^c^ calculated by Student’s *t*-test.

**Table 2 biosensors-12-00879-t002:** Determination of *cis* P-tau in real human serum samples collected from healthy and diseased subjects by using immunosensor.

Sample	*cis* P-tau Concentration (pM) *	RSD (%)
Healthy	0.02 ± 0.0011	5.5
MCI	0.05 ± 0.0025	5.0
Dementia-1	0.18 ± 0.0080	4.4
Dementia-2	2.0 ± 0.0800	4.0
Dementia-3	3.1 ± 0.1085	3.5

* Mean ± standard deviation (*n* = 3).

**Table 3 biosensors-12-00879-t003:** A comparison between this work and similar works on AD detection based on tau measurement.

Method	Analyte	LOD	Sample	Labeling	Reference
Electrochemical	Tau	0.2 μM	BSA	Label-free	[20]
Electrochemical	Tau	1000 pg/mL, 100,000 pg/mL	PBS, Serum	Label-free	[23]
Electrochemical	Tau	<1 pM	CSF	Label-free	[24]
Electrochemical	Tau-381	0.42 pM	Serum	Labeled	[21]
Photoelectrochemical	Tau-381	0.3 fM	Serum	Labeled	[69]
Electrochemical	full-length 2N4R tau	0.03 pM	Serum	Label-free	[17]
Impedimetric	Tau-441	0.091 pg/mL	Serum, CSF	Labeled	[70]
Electrochemical	P-tau-441	0.02 pM	PBS	Label-free	[71]
Electrochemical	cis pT231-tau	0.02 pM, 0.05 pM	PBS, CSF, Serum	Label-free	

## Data Availability

Data is contained within the article and Appendix A.

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
