# Peer review of "Earlier Detection of Alzheimer’s Disease Based on a Novel Biomarker *cis* P-tau by a Label-Free Electrochemical Immunosensor"

_biosensors, 2022, doi:10.3390/bios12100879_

Round 1

Reviewer 1 Report

1.  Introduction.  The writing occasionally comes across as cumbersome and with considerable grammatical errors (e.g., line 57: “none of these biomarkers are is…” and line 58 “none are is, line 54: “samples…” (needs comma here and in several other places (i.e., line 273) before “which”). There is inconsistency in use of abbreviations: Alzheimer’s Disease is abbreviated or shortened as AD, Alzheimer’s, or Alzheimer.  Sentence in lines 55-57 needs improvement and is confusing because the subsequent paragraph argues that cis p-tau is a known early biomarker.  Paragraph #3 explains that cis p-tau accumulation in neurons causes neuronal cell death in AD, but reference #39 seems to argue that antibody to extracellular cis p-tau can neutralize disease progression.  Paragraph #4 is peppered with excessively self-congratulatory phrases (“…high selectivity…”, “…which is patented as the only” …, “for the first time…”,  “great applicability…”), adding to the overall awkwardness in the Introduction.

2.  Materials and Methods.  Section 2.1 lacks complete descriptions (i.e., city at least) of the source companies.  Section 2.3 lacks information on the isotype, purification, and purity of the anti-tau mAb since references #61 and #62 are duplicate and incomplete at best. Section 2.3 indicates that BSA was used at 0.5mM, which would be ~33 mg/ml, a dubiously high concentration.   Is SPE = SPGE?  Section 2.6 (and Section 3.3) lacks detail on the concentration and source of human serum in which p-tau was diluted.  Re Section 2.7:  Presumably serum was derived from clotted blood, correct? How were were the human samples stored?  Section 2.8: what does 0.1M PBS mean? Does “without pre-treatment” mean undiluted?  The ELISA lacks details about the concentrations of materials, rationale for the bicarbonate treatment, and type/source of plates used for adsorption.

3.  Results, Table 1. What is its basis for being able to detect by ELISA cis p-tau adsorbed to plates non-specifically in the presence of a large excess of irrelevant CSF protein, especially since this assay compares quite favorably to the other solid phase system using electrochemical readout, in which cis p-tau is affinity captured from the same sources?

Reviewer 2 Report

Authors written and explained well in electrochemical part, real sample analysis and compared with the commercial ELISA. Only the Physical characterization is poor.  This manuscript can be accepted after resolving my following concerns-

Comments-
1. Author should include one comparison table for the performance with other recently reported works?

2. Are there any reason to use mAb instead of polyclonal Ab?

3. How did they confirm functionalization? Author should do some elemental analysis and functional group analysis eg XPS, RAMAN/FTIR.

4. How many sensors were used for measuring the whole dynamic range?

5. Fig 3a, how much Rct value was exhibited for each step? Authors should mention the individual values.  And why the second step Rct is lower than the bare electrode? seems controversial with CV curve? Please clarify it.

5. What is the novelty claim in this manuscript need to mention clearly?
6. Authors also  should mentioned about the label free based advantages?

Round 2

Reviewer 2 Report

Its better to add the dynamic range in the comparison table. 

Authors reflected all the concerns raised by me. Now It can be accepted in the current form.